# Enhanced bed shear stress and mixing in the tidal wake of an offshore wind turbine monopile

Martin J. Austin[1], Christopher A. Unsworth[1], Katrien J.J. Van Landeghem[1], and Ben J. Lincoln[1]

[1]Bangor University, School of Ocean Sciences, Menai Bridge, Anglesey, LL59 5AB, UK

**Correspondence:** Martin J. Austin (m.austin@bangor.ac.uk)

**Abstract.** Tidal flow past offshore wind farm (OWF) infrastructure generates a turbulent vortex wake. The wake is hypothesised to enhance seabed stress and water column turbulence mixing, and thereby affect seabed mobility, water column stratification, the transport of nutrients and oxygen, and result in ecological impact. We collect novel hydrodynamic data $40\,\mathrm{m}$ from an OWF monopile over a spring-neap cycle, and use high frequency velocity measurements to quantify turbulence. Outside of the wake we observe a classical depth-limited boundary layer, with strong turbulence production and dissipation forced by tidal shear at the seabed. Inside the wake, turbulence production, dissipation and stress are enhanced throughout the full water column and are maximised in the upper half of the water column, where they correspond to a strong mean velocity deficit. Our results show that the seabed drag coefficient is doubled from $C_d = 3.5 \times 10^{-3}$ to $7.8 \times 10^{-3}$, suggesting greater seabed mobility, and the eddy viscosity is increased by an order of magnitude indicating enhanced water column mixing. This research provides some valuable insight as OWFs expand into deeper seasonally stratified waters using both bottom-fixed and floating structures, where the addition of enhanced wake turbulence may have broad impacts as the additional mixing energy is added to regions with low rates of background mixing.

## 1 Introduction

There is growing interest in tidal flow past offshore wind turbine foundations and the impact of the associated wake on the environment through seabed sediment scour, water column mixing and the resultant ecological impacts (e.g., Chambel et al., 2024; Schultze et al., 2020; Dannheim et al., 2020). This is being driven by the planned massive expansion in offshore wind generation over the coming decade in the drive for Net Zero energy production. In northwest European waters (EU+UK) there were 36 GW of operational offshore wind in 2023, which is planned to increase to 110 GW by 2030 as large arrays of turbines are being installed in increasingly deep waters (TCE, 2022; European Commission, Directorate-General for Energy, 2023).

In a natural tidal flow, small-scale turbulent eddies generate strong velocity gradients (shear) close to the seabed that impart a high stress onto the bed. Moving away from the seabed, the scale of the turbulent eddies mediating the transfer of momentum increases, which reduces the velocity gradients and results in correspondingly smaller stresses; this leads to the law-of-the-wall benthic boundary layer. Turbine foundations add barotropic drag to the tidal flow, creating a deficit in the mean velocity and forming a vortex wake that generates additional turbulence at the scale of the monopile. In shallow waters the wake turbulence can directly impact the seabed and enhance the stresses, which may in turn modify seabed morphology (Couldrey et al., 2020),

sediment composition (McCarron et al., 2019) and impact benthic ecological communities (Damveld et al., 2018). The spatial and temporal persistence of the wake depends on the level of background turbulence, with wakes being efficiently eroded in highly turbulent (i.e. shallow and tidally energetic) regions (e.g., Eames et al., 2011).

In lower energy deeper waters where stratification may occur, and particularly with the transition to deep-draft floating foundations (e.g., Hywind Scotland 78 m draft spar buoy foundations (Equinor, 2023)), the combination of barotropic and baroclinic drag may significantly affect the water column by directly adding turbulent energy to the pycnocline. The addition of wake turbulence to these lower energy deeper environments will likely lead to more persistent regions of enhanced turbulent mixing (Dorrell et al., 2022); at the array scale, this may effect seasonal stratification (Schultze et al., 2020) and cause cumulative ecological impacts (Isaksson et al., 2023).

The aim of this paper is to investigate how the signature of the turbulent wake from an offshore wind turbine monopile differs from that of the background flow. Specific objectives are to quantify the changes to: (1) the rate of turbulence dissipation and production measured inside and outside the wake; (2) their vertical distribution through the water column; and (3) to assess the potential impact of the wake on seabed stress and water column mixing. We use field observations in a tidally energetic well-mixed environment and through the precise deployment of instruments 40 m from a fixed seabed monopile, measure natural background flows during the flood tide but sample directly within the wake during the ebb.

## 2    Location and Methods

### 2.1    Field Site

New observations were made in the western region of Liverpool Bay, UK, in the eastern Irish Sea during September 2022 (Fig. 1a, b). This shallow region experiences very large tides in the form of a semi-diurnal standing wave with springs ranges at Liverpool approaching 10 m and strong currents that frequently exceed 1 m.s$^{-1}$ which, away from the region of freshwater influence, maintain a well-mixed water column (e.g., Rippeth et al., 2001).

### 2.2    Monopile Wake Measurements

High resolution measurements were made $x = 41$ m northwest of a $D = 4.7$ m diameter monopile of the Rhyl Flats OWF in Liverpool Bay (53.38922N$^\circ$, 3.6866W$^\circ$); this equates to a distance $x/D$ of 8.7 (Fig. 1b, c). The location was selected so that during the flood tide it was upstream of the monopile and experienced natural background flows, whereas during the ebb it was downstream of the monopile within the wake. During the observations, the water depth varied between 12.5 m and 21 m on peak spring tides. The depth-averaged flow velocities confirm the rectilinear nature of the tidal flows (Fig. 2a). The dominant easterly component of the flood peaked at 0.8 m.s$^{-1}$; the ebb tides were weaker, reaching -0.6 m.s$^{-1}$, and show greater scatter as also observed at a nearby location by Unsworth et al. (2023). The cylinder Reynolds number $Re_d$,

$$Re_d = \frac{u_\infty d}{\nu} \tag{1}$$

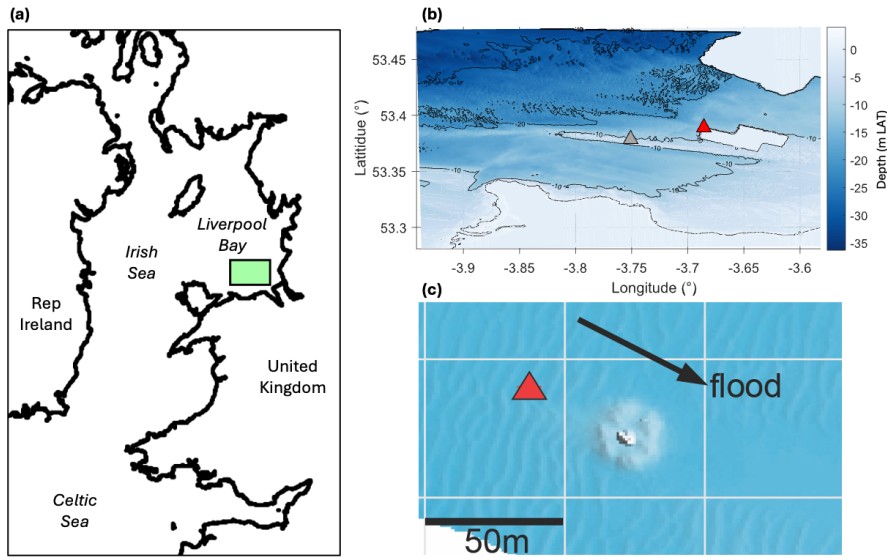

**Figure 1.** Field setting of the observations. (a) Outline map showing the location of the study site (green box) within Liverpool Bay in the eastern Irish Sea (b) Bathymetry plotted with depths as metres below Lowest Astronomical Tide (mLAT) showing the location of the Rhyl Flats OWF (UKHO, 2024) with the ADCP (red triangle) positioned at its north-west corner and the Constable Bank ADCP 3 km further to the south west away from the influence of the OWF (Unsworth et al., 2023).(c) Zoomed bathymetry showing the monopile and its surrounding rock scour protection, the ADCP location (red triangle), and indicating the flood tide direction.

where $u_\infty$ is the freestream flow velocity and $\nu$ the kinematic viscosity ($1.36 \times 10^{-6}$ m$^2$.s$^{-1}$), was $\mathcal{O}(10^4 - 10^6)$, for the range of observed depth-averaged flood tide velocities (Fig. 2b). The transition to the turbulent critical flow regime around a cylinder and the formation of an asymmetric Karman Vortex (KV) lee wake begins at $Re_d \gtrsim 2 \times 10^5$, which equates to a minimum velocity of 0.06 m.s$^{-1}$, and at $Re_d \gtrsim 5 \times 10^5$ a further transition to the super-critical flow regime and a symmetrical KV wake

occurs (e.g., Williamson, 1996). We therefore, expect to observe a turbulent symmetric KV wake in the lee of the monopile for the majority of the survey. The seabed was composed of rippled sand with a median grain size $d_{50} = 0.25$ mm determined by standard sieve analysis of multiple seabed grab samples. Following the method of van Rijn (1987), the seabed roughness height was $k_b = 0.122$ m, including both grain and bedform roughness elements. Rock scour protection is deployed to a diameter of $\sim 20$ m in the immediate area surrounding the base of the monopile.

Direct measurements of the wave field were obtained from the nearby Rhyl Flats wave buoy (53.38241N$^\circ$, 3.6062W$^\circ$; Channel Coastal Observatory (2022)). The wave height and period were typically less than $H_s = 1$ m and $T_z = 4$ s, respectively, but peaked at $H_s = 1.5$ m and $T_z = 4.5$ s, during $15 - 16$ Sep (Fig. 3d). In this shallow, tidally energetic region the water column

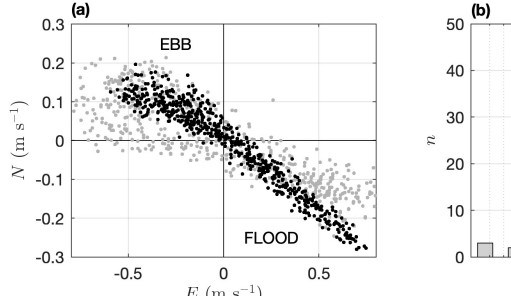

**Figure 2.** (a) Depth-averaged east and north tidal current velocities (black); also plotted (grey) are velocities from 3 km to the south west (grey triangle in Fig.1b), away from the influence of the OWF (Unsworth et al., 2023). (b) Histogram of cylinder Reynolds number $Re_d$, showing the transition to a turbulent boundary layer around the monopile and asymmetric KV wake at $Re_d \gtrsim 2 \times 10^5$ (dashed), and the super-critical transition to a symmetric wake at $Re_d \gtrsim 5 \times 10^5$ (dot-dash), (Williamson, 1996; Rodríguez et al., 2015).

generally remains well-mixed (cf. Rippeth et al., 2001; Simpson et al., 2002). No profiles of water temperatures were recorded, but timeseries of surface and bottom temperatures recorded nearby during 2018 indicate that thermal stratification typically
only occurs during peak neap tide conditions during the months of June – July.

### 2.3  Observational Setup

A Nortek Signature 1000 5-beam acoustic Doppler current profiler (ADCP) was installed on a bed-mounted frame and profiled the water column within 44 bins of 0.5 m thickness; the centre of the lowest bin was at $z = 0.8$ m above the seabed. The instrument heading was 220°, with the plane of beams 2 and 4 aligned with the streamwise current; the pitch and roll of the
instrument on the seabed were -0.5° and 5.3°, respectively. The ADCP recorded along-beam velocities with a ping rate of 8 Hz and operated in burst mode collecting 2048 samples at the start of each hour. The Doppler noise level was reported by the instrument as $\sigma_N = 0.016$ m.s$^{-1}$ as configured for this deployment.

### 2.4  Turbulence Metrics

The dissipation rate ($\varepsilon_5$) of turbulent kinetic energy (TKE) was estimated from the along-beam velocities of the vertical fifth
beam of the ADCP using the Structure Function method (Wiles et al., 2006). In the presence of surface gravity waves, the vertical gradient in wave orbital velocity will bias the along-beam velocities, and we use the modified approach of Scannell et al. (2017) to remove this bias (Equation A3).

The Reynolds stresses were estimated using the variance method (Lu and Lueck, 1999; Rippeth et al., 2003), extended by Dewey and Stringer (2007) for a five-beam ADCP and including non-zero pitch and roll. The fifth beam provides an
independent measure of the vertical velocity $w$, and allows the stress components to be estimated with a reduced bias because

the approximation due to the orthogonal beam is removed. The coordinate system was rotated to provide the streamwise ($\overline{u'w'}$), cross-stream ($\overline{v'w'}$) and vertical stresses ($\overline{w'w'}$).

In a well-mixed tidally dominated environment, TKE production $P$ is primarily through energy transfer from the mean flow to turbulence via shear at the seabed; the buoyancy term can be neglected. $P$ was estimated from the Reynolds stresses and the mean velocity vertical shear (e.g., Rippeth et al., 2003).

## 3 Results

### 3.1 Variation in Rates of Turbulence Dissipation and Production

The principal environmental parameters measured adjacent to the monopile are shown in Fig. 3. The profile of the mean streamwise velocity $\overline{U_x}$ (Fig. 3a) indicates that the semi-diurnal tidal wave is standing and exhibits strong semi-diurnal and spring-neap variation. The largest velocities are observed on the peak spring tides ($+0.8$ m.s$^{-1}$; -0.6 m.s$^{-1}$), reducing to $\pm 0.35$ m.s$^{-1}$ during neaps.

The TKE dissipation rate $\varepsilon_5$ (Fig. 3b) has both a spring-neap and a pronounced quarter diurnal variation. During the flood, maximum values of $\varepsilon_5$ are -4.5 log$_{10}$(W.kg$^{-1}$) during springs and -5 log$_{10}$(W.kg$^{-1}$) during neaps. In contrast, maximum $\varepsilon_5$ during the ebb increases by almost an order of magnitude during springs (-3.7 log$_{10}$(W.kg$^{-1}$)) and half an order of magnitude during neaps. During spring tides, significant levels of $\varepsilon_5$ is observed to occur through the full depth of the water column.

The TKE production rate $P$ (Fig. 3c) follows a similar temporal pattern to $\varepsilon_5$. During both phases of the tide, strong production (-4.5 log$_{10}$(W.kg$^{-1}$)) consistent with shear at the seabed extends up to $z \approx 3$ m above the bed, but during the ebb, an additional vertical band of strong $P$ is present through the full water column. At several times throughout the survey (e.g., 15 – 19 Sep), intense regions ($>$-4 log$_{10}$(W.kg$^{-1}$)) of near-surface $P$ occur, which propagate downwards into the water column to $z = 10$ m. Comparing these periods to the surface waves recorded at the nearby wave buoy (Fig. 3d), steep waves ($H_s = 1.5$ m, $T_z = 4$ s) are likely injecting TKE directly into the surface layers and contributing to $P$. This wave contamination is not observed in $\varepsilon_5$ due to our use of the modified structure function (Equation A3).

### 3.2 Wake-modified Vertical Distribution of Turbulence

To provide further insight into the distribution of turbulence in the water column and to identify the balance between seabed and water column processes, we explore three spring tidal cycles in greater detail and compare with theoretical scaling. Fig. 4a highlights the significant flood-ebb asymmetry in the strength and spatial distribution of $\varepsilon_5$. During the upstream (flood) phase, $\varepsilon_5$ appears to be forced at the seabed and propagates vertically toward the surface as the flow accelerates before weakening prior to high water slack. In contrast, during the downstream (ebb) phase, high $\varepsilon_5$ persists through the full water depth with maximum values observed in the upper water column. This flood ebb asymmetry is unlikely attributable to tidal straining or convection, because of the well-mixed nature of the environment. In Fig. 4b we integrate $\varepsilon_5$ across 2 m-thick bands of the water column close to the seabed and near the sea surface, and through the full water column. All three display a quarter-diurnal variation

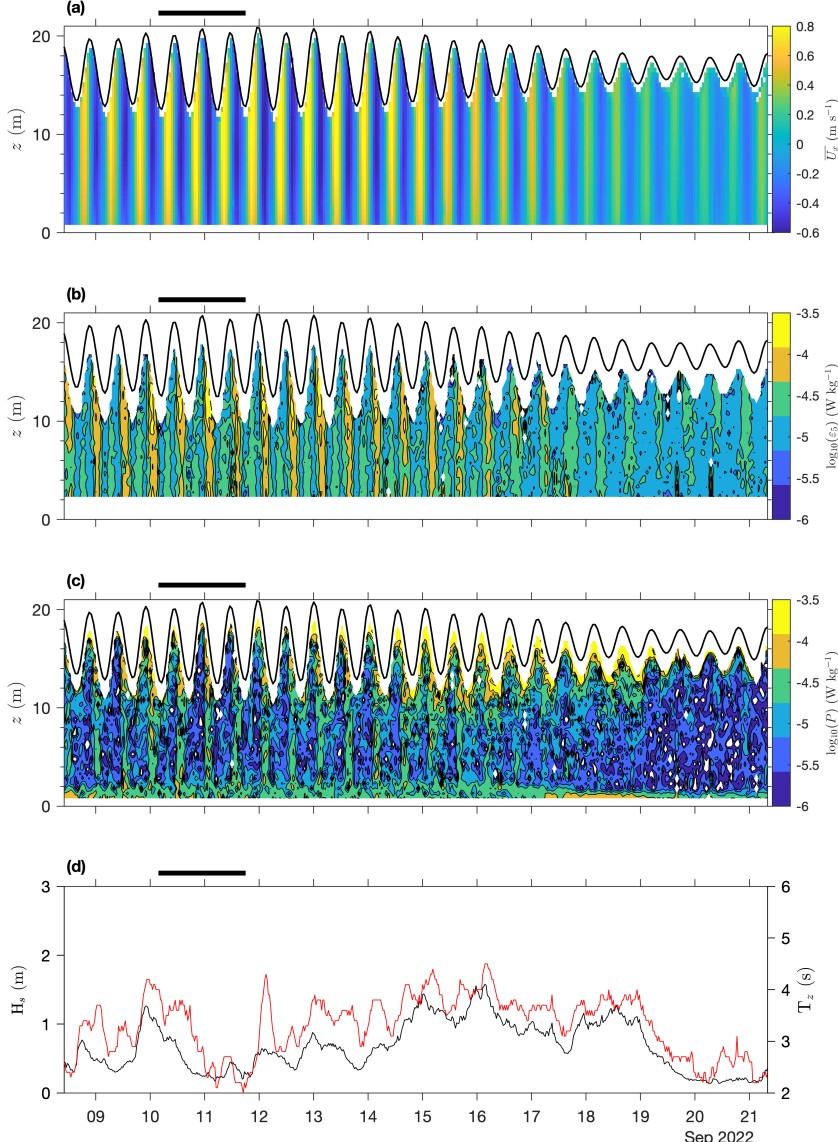

**Figure 3.** Overview of observations during the survey. (a) Streamwise mean velocity $\overline{U_x}$; (b) turbulence dissipation rate $\varepsilon$; (c) turbulence production $P$; and (d) significant wave height $H_s$ (black) and period $T_z$ (red). The water surface elevation is shown by the black line in (a–c). The white band at $z < 2$ m in (b) is due to the minimum of four bins for the regression of $D(z, r)$ against $r^{2/3}$ in Equation A3. The horizontal black bar above each panel highlights the period shown in Fig. 4.

in $\varepsilon_5$, but contrasting behaviour is seen at the seabed and near surface during upstream and downstream phases. During the upstream phase, $\varepsilon_5$ at the seabed exceeds that near the surface, but during the downstream phase this pattern is reversed and the difference between them is approximately twice as large. Integrating through the full depth of the water column, $\varepsilon_5$ is approaching an order of magnitude greater during the downstream phase. We do note that $\varepsilon_5$ is possibly under-estimated at the seabed, since due to the requirement of a minimum of four bins for the regression in (Equation A3), the lower 2 m is excluded.

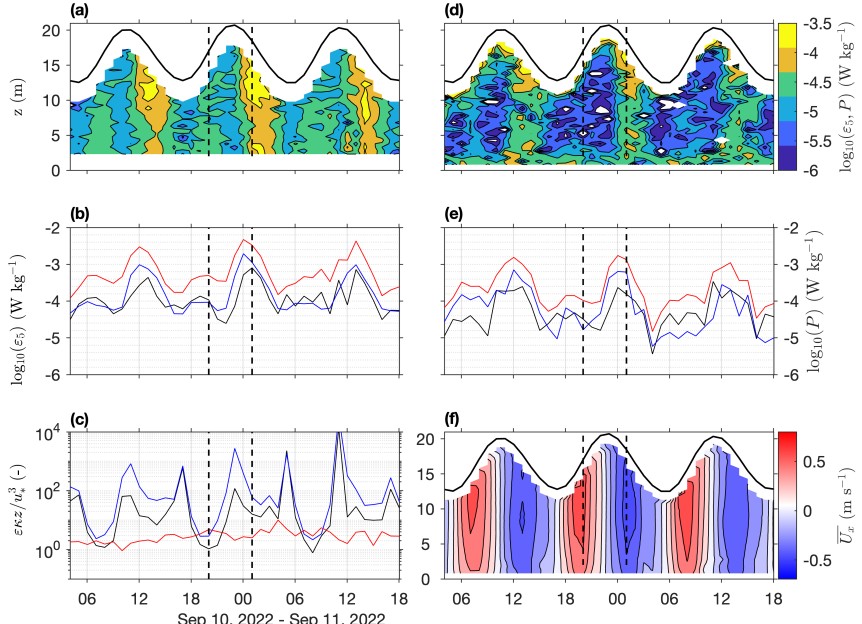

**Figure 4.** Vertical water column distribution of turbulence dissipation and production rate. (a) dissipation rate $\varepsilon_5$; (b) seabed (black), near-surface (blue) and total (red) dissipation rate $\varepsilon_5$; (c) similarity scaling of dissipation rate $\varepsilon_5$ at seabed (black), near-surface (blue), and total observed ratio $\varepsilon_5/P$ (red); (d) turbulence production $P$; (e) seabed (black), near-surface (blue) and total (red) turbulence production $P$; and (f) mean streamwise velocity $\overline{U_x}$. Vertical dashed lines highlight equivalent mid-tide upstream (flood) and downstream (ebb) periods.

Following similarity theory, a local balance should exist between $\varepsilon$ and $P$ based on a constant stress relationship, with the local production of turbulence by tidal shear at the seabed. This leads to a scaling factor $\varepsilon_s = -u_*^3/\kappa z$, where $u_*$ is the friction velocity, calculated as $u_* = (\tau_b/\rho)^{1/2}$ for bottom shear stress $\tau_b$ and seawater density $\rho = 1027$ kg.m$^{-3}$; $\kappa$ is the von Karman constant (= 0.41). Here we use the method of Soulsby and Clarke (2005) to compute the current only $\tau_{b,c}$ with the measured bottom roughness length-scale $k_b = 0.122$ m, accounting for both bedform and skin friction roughness. During the upstream phase (Fig. 4c) $\varepsilon_5$ close to the seabed is in almost perfect balance with similarity theory, but at the surface is approximately 20% higher. During downstream conditions, $\varepsilon_5$ is one and 1.5 orders of magnitude greater than predicted at the seabed and near-surface, respectively, and peaks prior to the occurrence of the strongest flows (mid-ebb); this likely reflects the addition

of wake turbulence to the water column. The similarity scaling breaks down at slack water as $\varepsilon_5$ is maintained despite the cessation of local production from the mean flow.

A flood-ebb asymmetry of $P$ is clearly shown in Fig. 4d, which shows a consistent near-bed TKE production on both upstream and downstream flow phases and a strong production throughout the full water column during the downstream phase. Fig. 4c integrates $P$ into vertical zones and also shows the asymmetric flood/ebb behaviour, with greater downstream TKE production ($P$ =-3.5 $\log_{10}$(W.kg$^{-1}$)) compared to during upstream conditions ($P$ =-4.5 $\log_{10}$(W.kg$^{-1}$)). The difference between seabed and near-surface $P$ is more variable, but overall we observe a trend of greater seabed production during upstream flows and greater near-surface production during downstream flows. When integrated through the full water column, $P$ and $\varepsilon_5$ are in close agreement in magnitude and trend, and the observed budget $\varepsilon_5/P$ remains within an order of magnitude (Fig. 4c, red line).

The mean streamwise velocity $\overline{U_x}$ highlights the clear differences between the background upstream flood-tide flows and the downstream wake-effected ebb (Fig. 4f). During the upstream phase the characteristic tidal benthic boundary layer structure is evident with the strongest flows at the surface, which reduce towards the bed developing shear. During the downstream phase the velocity magnitude is reduced by approximately 35 % with the strongest flows occurring at mid-depth and shear developing close to the bed. In the upper water column, a significant velocity deficit indicates the presence of the monopile wake and is in-phase with the peaks in $\varepsilon_5$ and $P$.

### 3.3   Enhanced Reynolds Stress, Seabed Drag and Mixing

To provide insight into the effect of the monopile wake through the water column we compute ensemble-averaged vertical profiles of mean velocity, stress and TKE dissipation rate (Fig. 5). Vertical profiles at mid-flood and mid-ebb, when flow accelerations are expected to be minimal, were extracted from the seven spring tides between 9 and 13 Sep and geometrically averaged in $z/h$-space to account for tidal changes in water depth. During the upstream phase (flood), the mean velocity profile displays a logarithmic form and extends the full thickness of the water column as expected in a system dominated by tidal shear at the seabed (Fig. 5a). The maximum values of the stress occur at or just above the seabed (Fig. 5b) and tend towards zero moving away from the seabed (Rippeth et al., 2002); an increase in stress is observed for $z/h > 0.65$, which may be due to surface waves. Following Rippeth et al. (2003), for our observed wave amplitude and period of $\sim$0.5 m and $\sim$4 s, respectively, we may expect a bias in the stress term close to the surface of $\sim \mathcal{O}(0.5)$ N.m$^{-2}$ decreasing with depth to $\sim \mathcal{O}(0.05)$ N.m$^{-2}$ at the bed, which is in close agreement with our observations.

During the downstream wake-affected ebb tide, a strong velocity deficit is present above $z/h > 0.5$ (Fig. 5a), indicative of the core of the wake. The stress increases from minimum values near the seabed to extreme values at the surface, but there is also a region of highly variable stress at $0.1 \leq z/h \leq 0.3$ (Fig. 5b). It is noteworthy that for equivalent stages of the tide, whilst the downstream velocities are significantly weaker than during the upstream phase, the stresses are up to six times larger; this far exceeds the potential bias in the stress terms due to surface waves.

The profiles of $\varepsilon_5$ display a similar upstream-downstream asymmetry to the stresses (Fig. 5c). Peak $\varepsilon_5$ values occur during the downstream phase, with extreme values (-3.5 $\log_{10}$(W.kg$^{-1}$)) at $z/h = 0.25$ and above mid-depth $z/h > 0.50$. Again,

despite the stronger upstream velocities, $\varepsilon_5$ is almost an order of magnitude weaker during the upstream phase, which display
maximum values close to the seabed and decrease linearly to the surface.

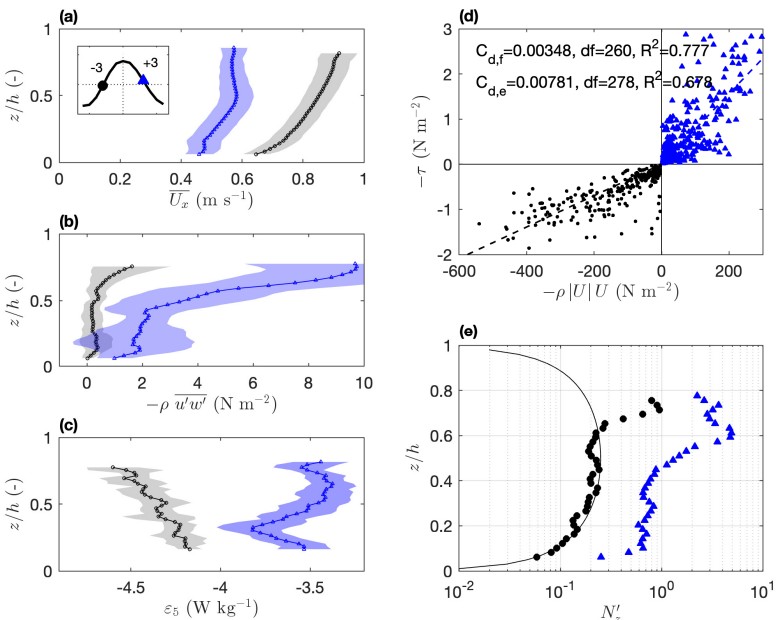

**Figure 5.** Summary of stress and drag. Peak upstream (black) and downstream (blue) ensemble-averaged (a) mean streamwise velocity profiles $\overline{U}_x$, where the inset shows the peak flood and ebb times in the tidal cycle; (b) Reynolds stress $-\rho\overline{u'w'}$; and (c) dissipation rate $\varepsilon_5$. Shaded regions plot $\pm$ standard deviation. (d) Comparison of the Reynolds stress estimates with the quadratic drag law during upstream (black) and downstream (blue) periods. The Reynolds stress averaged over the four bins closest to the bed ($z = 0.8 - 2.3$ m) is plotted against $-\rho|U|U$, where $U$ is the mean current speed at z = 1 m, and the drag coefficient $C_d$ is evaluated by linear fitting with the respective $R^2$ and degrees of freedom ($df$). (e) Non-dimensional eddy viscosity $N_z'$ (where $N_z' = N_z/\kappa u_* h$) for ensemble-averaged upstream and downstream phases compared to the steady-flow model (solid line).

Stress at the seabed is often parameterised using the drag coefficient $C_d$, and the quadratic stress law $\tau_b = \rho C_d \overline{U}^2$, to relate the mean flow speed $\overline{U}$ defined at some elevation above the seabed (frequently $z = 1$ m, e.g., Soulsby, 1998) directly to the turbulent stress. We compare the total shear stress $\tau = -\rho\sqrt{\overline{u'w'}^2 + \overline{v'w'}^2}$, averaged over the lowest 2 m of the water column, with the quadratic stress law evaluated at $z = 1$ m above the bed (Fig. 5d). Upstream (flood) and downstream (ebb) data are
independently fitted to a straight line (both with $R^2 \sim 0.7$ and $\sim 270$ degrees of freedom) with significantly different slopes. This yields an downstream ebb drag coefficient of $C_{d,e} = 7.8 \times 10^{-3}$, twice that observed during the upstream flood phase ($C_{d,f} = 3.5 \times 10^{-3}$).

The eddy viscosity $N_z$ relates the vertical velocity gradients to the horizontal stress and reflects the role of turbulent eddies in mediating the transfer of momentum. If we assume a steady flow model where the velocity shear is described by $\partial u/\partial z =$

$u_*/\kappa z$, and the stress decreases linearly from its maximum value at the bed $\tau_b = -\rho u_*^2$, to zero at the surface, $N_z$ takes a parabolic form

$$N_z = \kappa u_* z \left(1 - \frac{z}{h}\right) \tag{2}$$

(Rippeth et al., 2002). We compare $N_z$ computed for the ensemble-averaged peak upstream and downstream tidal phases in Fig. 5e. During the upstream phase, although there is some significant deviation from the steady-flow model approaching the sea

surface for $z/h > 0.6$, the observations are consistent with the model in both trend and magnitude. However, during the wake-effected ebb tide, $N_z$ is almost an order of magnitude larger than the steady-flow model and although approximating a parabolic form at $z/h < 0.35$, it rapidly increases in the upper water column between $z/h = 0.35$ and $z/h = 0.6$; this corresponds to the region where shear, stress and $\varepsilon_5$ are observed to increase for the wake-affected flow in Fig. 5a–c and suggests that the monopile drives enhanced vertical mixing through the water column.

## 4    Discussion

### 4.1    Wake Enhanced Turbulence

The present study utilises high-frequency 5-beam ADCP data to provide turbulence metrics over multiple tidal cycles within the tidal wake 40 m downstream of an offshore wind turbine monopile. Observations of the TKE dissipation rate and Reynolds stresses have been combined with the mean velocity shear to provide new insight into the changes to the vertical structure of the

water column inside and outside of the wake. Upstream of the monopile, these parameters generally accord to a classical depth-limited law-of-the-wall boundary layer with peak values of mean velocity shear and stress close to the bed, and correspondingly high rates of TKE production and dissipation that are balanced at the seabed (Fig. 4). The reduction in $\varepsilon_5$ moving away from the seabed indicates that the Kolmogorov length scale $(\nu^3/\varepsilon)^{1/4}$, and thus the size of the energy containing eddies, is increasing as a function of $\kappa z$. Downstream of the monopile within the wake, a strong velocity deficit forms in the upper half of the water

column, and the production and dissipation rates of TKE deviate from theoretical similarity scaling and increase throughout the water column above the region of tidal shear in the bottom boundary layer (Fig. 4). We have further tested the impact of these wake-generated variations on the seabed and on the vertical water column mixing and show that the seabed drag coefficient and the eddy viscosity, respectively, are enhanced.

     The monopile spans the full depth of the water column and extracts momentum from the mean flow, which is transferred

to TKE at a (large) scale similar to that of the monopile itself (4.7 m); this leads to the observed velocity deficit in the wake that advects downstream in a self-similar manner. In the present ebb tide data we observe that the velocity deficit is largely eliminated in the lower half of the water column ($z/h \leq 0.5$). This indicates that the high levels of shear and stress measured close to the bed, which result from the combination of natural small-scale eddy interactions and the additional TKE introduced by the monopile itself (Fig. 5), provide an important restorative mechanism for the mean flow by eroding the wake as it is

advected downstream (Eames et al., 2011). Further support for this argument is provided by the high rate of TKE dissipation $\varepsilon_5$ also observed close to the bed.

Higher in the water column the velocity deficit is maintained due to a reduction in the efficiency of the ambient flow to erode the wake. First, the difference between the ambient flow and the velocity deficit is much greater due to the reduction in frictional drag from the seabed, and so the work required to restore the mean flow near the surface is therefore much greater. Second, we expect a decrease in the ambient shear and stress as the eddy length scale increases in proportion to $\kappa z$ to also reduce the rate at which the wake is eroded. The combination of these factors plus the continued addition of TKE from the deficit (Scott et al., 2023) leads to the maintenance of a region of elevated shear above $z/h = 0.5$ that is associated with very high stress, and $\varepsilon_5$, which is maximised at $0.5 \leq z/h \leq 0.7$. Overall, this combination of seabed and water column factors suggests that wakes should persist for longer in environments with lower rates of background turbulence mixing, for example deeper, less tidally energetic regions.

## 4.2 Implications of Enhanced Seabed Drag

We show that the high levels of stress observed in the wake directly increase the drag coefficient at the seabed. Outside of the wake $C_{d,f} = 3.5 \times 10^{-3}$, which is comparable to standard values reported for sandy and sand-gravel seabed mixtures (e.g., Soulsby, 1998), but this is more than doubled to $C_{d,e} = 7.8 \times 10^{-3}$ within the wake. Unless this wake-enhanced transfer of momentum from the mean flow to turbulence is accounted for, shelf-scale models (e.g., Telemac, FVCOM) that generally use the quadratic stress law as the default method of estimating bed shear stress will under predict seabed stress even if they include monopiles as point drag sources (e.g., Rennau et al., 2012; Christiansen et al., 2023). The elevated $C_d$ within the wake can be directly linked to enhanced sediment transport and observations of morphological change in the lee of monopiles. Wakes can modify the seabed, flattening bedforms by forcing a change towards an upper-stage plane bed regime as observed at Rhyl Flats OWF downstream of the monopile (Fig. 1c), and in extreme cases causing seabed scour, or can force a live-bed regime to transition towards one that supports bedform development (Couldrey et al., 2020). At the array-scale, wake formation will increase the spatial variability of the seabed morphology, change grain size mixtures, and likely promote greater seabed heterogeneity; this can impact the make-up and functionality of the benthic ecosystem (van der Kooij et al., 2008; Gates et al., 2019) and feed up to higher trophic levels.

## 4.3 Implications of Enhanced Eddy Viscosity

The wake enhances water column mixing by modifying the local momentum flux, and the eddy viscosity provides a useful concept for estimating wake mixing efficiency and persistence by directly relating the mean flow gradients to the turbulent stresses. The magnitude of the eddy viscosity $N_z$ is controlled by the interplay between the mean flow shear, the rate of recovery of the velocity deficit, and the dissipation rate $\varepsilon_5$. In the present data, the ADCP is located at $x/D = 8.7$, which is sufficiently far downstream of the monopile that the velocity deficit is starting to recover and close to the seabed, where strong frictional drag and high $\varepsilon_5$ occur, $N_z$ is reduced. However, higher in the water column where the velocity deficit remains significant the rate of transfer of momentum to TKE remains high relative to $\varepsilon_5$, and strong Reynolds stresses are still generated maximising $N_z$ (Fig. 5). As the velocity deficit continues to weaken moving downstream we would expect $N_z$ to decrease as $\varepsilon_5$ drives the decay of the Reynolds stresses (Scott et al., 2023). The complex distribution of $N_z$ through the wake-effected water column has

significant implications for the inclusion of OWF turbulent wakes in large-scale shelf-sea models. The turbulence is sub-grid scale in these models, and the mixing processes are passed to some form of generic turbulence closure model, which attempts to balance the production and dissipation of TKE and return this as an $N_z$. In light of the many length scales introduced by the structures, robust datasets and fully resolved model simulations are first required to provide further insight into the physics of flow past structures, particularly in stratified regions.

## 4.4 Broader Consequences of Enhanced Mixing

The observed order of magnitude increase in mixing caused by the monopile is likely to have significant implications for the transition towards floating OWF in deeper seasonally stratified waters (Dorrell et al., 2022). In these regions, background mixing rates are low because the pycnocline isolates the surface from the bottom waters, which limits the flux of nutrients and oxygen that drive primary productivity (Sharples et al., 2001). The addition of the high level of anthropogenic mixing
caused by tidal flow past OWF structures to the stratified regions will result in a large relative increase in mixing compared to the background levels, potentially affecting the timing of the onset and breakdown of seasonal stratification (Rippeth, 2005), the strength of the stratification (Schultze et al., 2020), and the mixed-layer depth (Pearson et al., 2015), combined this could result in significant ecological impacts spanning multiple trophic levels from plankton to fish to top predators (Trifonova and Scott, 2023). However, wake mixing dynamics in stratified fluid flows are presently poorly understood and high fidelity large
eddy simulation modelling is necessary to capture the baroclinic and advective processes that are required in order to produce a generic wake parameterisation for broad implementation.

## 5 Conclusions

High-frequency velocity measurements are used to quantify the turbulent vortex wake 40 m downstream of an OWF monopile in a tidally energetic environment. We take advantage of the rectilinear tidal flow to compare the natural background upstream
flow during the flood tides with the wake-affected downstream flow during the ebb. The rates of turbulence production and dissipation are driven by tidal shear at the seabed during background flow conditions and reach -4.5 $\log_{10}(\mathrm{W.kg^{-1}})$. Within the wake a strong mean flow velocity deficit develops, which drives enhanced dissipation and production of -3.5 $\log_{10}(\mathrm{W.kg^{-1}})$ through the full depth of the water column. Reynolds stresses are also enhanced within the wake. This doubles the seabed drag coefficient from $C_d = 3.5 \times 10^{-3}$ to $C_d = 7.8 \times 10^{-3}$ and implies that seabed mobility will increase resulting in greater
seabed heterogeneity. Enhanced stresses also increase the eddy viscosity by an order of magnitude, which will drive greater vertical water column mixing. These results provide useful insight as OWF developments progress into deeper often seasonally stratified waters, where the addition of extra turbulent energy into the water column may alter the present delicate balances and result in widespread ecosystem impact. Future work should focus on the generic parameterisation of wake turbulence into the shelf-scale numerical models required for planning and impact mitigation purposes.

*Data availability.* The Rhyl Flats wave buoy data may be freely downloaded from the Channel Coastal Observatory (https://coastalmonitoring. org/realtimedata/?chart=100&tab=download&disp_option=). The raw ADCP data is available at https://zenodo.org/doi/10.5281/zenodo. 12530956.

## Appendix A: TKE Dissipation Rate

The TKE dissipation rate $\varepsilon$ was derived from the ADCP velocity measurements using the Structure Function method (Wiles
et al., 2006). Along-beam velocity components $b(z)$ from each beam are used to estimate the second-order structure function:

$$D(z,r) = \overline{\left( b'(z) - b'(z+r) \right)^2} \tag{A1}$$

where the overbar is the burst-average, $b' = b(z) - \bar{b}(z)$ is the fluctuating component of velocity at position $z$ along the beam and $D(z,r)$ is the mean-square of the velocity fluctuation difference between two points separated by distance $r$, based on multiples of the bin size. The maximum separation distance $r_{max}$ was set as 3 m. For isotropic turbulence, the structure function $D(z,r)$
is related to $\varepsilon$ by:

$$D(z,r) = C_2 \varepsilon^{2/3} r^{2/3}, \tag{A2}$$

where $C_2$ is a constant (=2.0). In the presence of surface gravity waves, the vertical gradient in wave orbital velocity will bias the along-beam velocities. We use the modified approach of Scannell et al. (2017) to remove this bias, using a least-squares fit to generate a linear model:

$$D(z,r) = a_0 + a_1 r^{2/3} + a_3 (r^{2/3})^3 \tag{A3}$$

where $a3$ contains the contribution to $D(z,r)$ from the waves, $a_0 = 2\sigma_b^2$ provides an estimate of the instrument noise, and the gradient $a_1$ is used to derive $\varepsilon$ via:

$$\varepsilon = \left( \frac{a_1}{C_2} \right)^{2/3}. \tag{A4}$$

Here, we estimate the dissipation rate from the vertical fifth beam of the ADCP ($\varepsilon_5$).

*Author contributions.* All authors designed the experiment. MJA and CAU performed the initial data post-processing, MJA undertook the analysis and visualisation of the data, and BJL helped with the structure function analysis. MJA prepared the initial manuscript and all authors contributed to its development.

*Competing interests.* The authors declare no competing interests.

*Acknowledgements.* This work was funded by the NERC ECOWind-ACCELERATE project (NE/X008886/1) awarded to KJJVL and MJA. MJA and BJL also acknowledge additional support from the EPSRC project Measurable metrics for characterisation of large-scale turbulent structures in tidal races for the marine tidal energy industry (EP/R000611/1) and the Smart Efficient Energy Centre (SEEC) funded by the Welsh European Funding Office (WEFO) as part of the European Regional Development Fund (ERDF). We would like to thank Ben Powell, Pete Hughes and Aled Owen for performing the instrument deployment and recovery. Tom Rippeth provided useful discussion on the analysis and manuscript.

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
