# Peer review of "Enhanced bed shear stress and mixing in the tidal wake of an offshore wind turbine monopile"

_EGUsphere, 2024_

## Referee Comment (RC1)

Figure 1:

a)

[Figure]

UK

Ireland

b)

c)

flood

— 0

— 5

...

— −40

Figure 2:

[Figure]

a)

N

E

b)

n

Re_critical

Red

---

## Referee Comment (RC2)

**Review of paper "Enhanced bed shear stress and mixing in the near wake of an offshore wind turbine monopile"**

Summary:

The aim of the paper is to investigate how the signature of the turbulent wake from an offshore wind turbine monopile differs from that of the background flow. They used ADCP high frequency flow analysis, in a rectilinear tidal flow and in and outside from the wake of a pile, and masured the average velocity, the Turbulent Kinetic Energy and dissipation, and analyse and compare vertical profiles for both ebb and flood tidal periods.

Vertical profiles upstream or downstream of the pile show an increase in turbulence downstream the turbine. The Seabed Drag Coefficient increased from $3.5 \times 10^{-3}$ to $7.8 \times 10^{-3}$ within the wake, indicating greater seabed mobility, and eddy viscosity is increased by an order of magnitude, suggesting enhanced water column mixing.

Implications of the monopile wake are an alteration of the vertical structure of the water column and an increase in turbulence and stress alongside an increase in column mixing that could impact seasonal stratification and have an ecological impact. Change in seabed mobility can imply changes in seabed morphology. They recommend considering the wake-induced turbulence in Environmental Impact Assessment for Offshore Wind Farm and introducing accurate calculation of bed shear stress within the wake in numerical model such as Telemac or FVCOM.

The paper is well presented, concise and features informative figures. I have only minor comments, most of them being typo or a comment that I hope may help the reader comprehending the paper.

Minor comments:

Page 2, section 2.1. Is there any scour protection at the monopile and where Observations were measured? Or did the scour reach an equilibrium stage? How homogeneous is the grain size over the area?

Page 2, line 54: Please indicate the formulation you have used?

Page 2, line 57: Please refer to Figure 2d or say full date/add the year (2022).

Page 2, line 57: I would not use a comma after energetic

Page 4, line 59: do you know if this was a one-off event in 2018? Or stratification pattern is always like that over the period/tidal cycle indicted?

Page 4, line 65: was this reported from the Device manufacturer?

Page 4, line 67 : TKE in full please for the time: Turbulent Kinetic Energy (TKE).

Page 4, line 81: The parameters were measured by the ADCP, not observed. And those measurement are observations. Please re-phrase.

Page 4, line 78: In Figure 3a, I read +0.8 m/s; -0.6 m/s. Not +/-0.8 m/s

Page 5, caption Figure 2: Please add "equation" before (A3)

Page 6, line 117: no dash in "breaks-down", breaks is the verb and down is an adverb.

Page 8, line 121-122: Can you please indicate to which Figure those comment refer to (3d or 3e?).

Page 8, line 124-125: Similar to previous comment, are you referring to Figure 3b-3e?

---

## Author Comment (AC1)

**Review of paper "Enhanced bed shear stress and mixing in the near wake of an offshore wind turbine monopile"**

**Summary:**
The aim of the paper is to investigate how the signature of the turbulent wake from an offshore wind turbine monopile differs from that of the background flow. They used ADCP high frequency flow analysis, in a rectilinear tidal flow and in and outside from the wake of a pile, and measured the average velocity, the Turbulent Kinetic Energy and dissipation, and analyse and compare vertical profiles for both ebb and flood tidal periods.
Vertical profiles upstream or downstream of the pile show an increase in turbulence downstream the turbine. The Seabed Drag Coefficient increased from $3.5 \times 10^{-3}$ to $7.8 \times 10^{-3}$ within the wake, indicating greater seabed mobility, and eddy viscosity is increased by an order of magnitude, suggesting enhanced water column mixing. Implications of the monopile wake are an alteration of the vertical structure of the water column and an increase in turbulence and stress alongside an increase in column mixing that could impact seasonal stratification and have an ecological impact. Change in seabed mobility can imply changes in seabed morphology. They recommend considering the wake-induced turbulence in Environmental Impact Assessment for Offshore Wind Farm and introducing accurate calculation of bed shear stress within the wake in numerical model such as Telemac or FVCOM.

The paper is well presented, concise and features informative figures. I have only minor comments, most of them being typo or a comment that I hope may help the reader comprehending the paper.

We thank the reviewer for their supportive comments on our manuscript and below we detail how we have addressed their minor comments within the revised manuscript. We hope that these changes improve the clarity of the paper for the reader.

**Minor comments:**
Page 2, section 2.1. Is there any scour protection at the monopile and where observations were measured? Or did the scour reach an equilibrium stage? How homogeneous is the grain size over the area? The base of the monopile is surround by rock armour scour protection to a diameter of approximately 20 m; this is evident in the inset pane of Figure 1, and we have noted it in the figure caption and the main body of text at line 64.

No scour is evident in the multibeam echo sounder data from upstream of the monopile, but downstream (in the direction of the flood tide) the bedforms are washed-out indicating the action of the wake at the seabed. Our stated $d_{50}$ = 0.25 mm grain size (line 61) was derived from the analysis of multiple sediment grab samples collected around the Rhyl Flats region, which is largely homogenous. We have modified the text to clarify these points at line 61 – 64:

*"The seabed was composed of rippled sand with a median grain size d_{50}= 0.25 mm determined by standard sieve analysis of multiple seabed grab samples. Following the method of \citet{vanRijn1987MathematicalMO}, the seabed roughness height was k_b= 0.122*

*m, including both grain and bedform roughness elements. Rock scour protection is deployed to a diameter of ~20 m in the immediate area surrounding the base of the monopile."*

Page 2, line 54: Please indicate the formulation you have used? We have added this to the text: van Rijn (1987).

Page 2, line 57: Please refer to Figure 2d or say full date/add the year (2022). Done.

Page 2, line 57: I would not use a comma after energetic. Removed.

Page 4, line 59: do you know if this was a one-off event in 2018? Or stratification pattern is always like that over the period/tidal cycle indicted? This is the regular pattern that is observed for this region of Liverpool Bay. We have cited Rippeth et al. (2001) and Simpson et al. (2002) on line 68 which provide detailed oceanographic background to this region.

Page 4, line 65: was this reported from the Device manufacturer? This is the value reported by the instrument manufacturer's software based upon the configuration settings of the instrument as deployed. We have clarified this in the text. *"The Doppler noise level was reported by the instrument as \sigma_N= 0.016 m s^{-1} as configured for this deployment."*

Page 4, line 67: TKE in full please for the time: Turbulent Kinetic Energy (TKE). Done.

Page 4, line 81: The parameters were measured by the ADCP, not observed. And those measurement are observations. Please re-phrase. We have re-phrased this statement.

Page 4, line 78: In Figure 3a, I read +0.8 m/s; -0.6 m/s. Not +/-0.8 m/s. The text has been updated to reflect this.

Page 5, caption Figure 2: Please add "equation" before (A3). Done.

Page 6, line 117: no dash in "breaks-down", breaks is the verb and down is an adverb. Changed.

Page 8, line 121-122: Can you please indicate to which Figure those comment refer to (3d or 3e?). We have updated the text throughout this section to be more explicit as to which figure sub-panel we were referring to.

Page 8, line 124-125: Similar to previous comment, are you referring to Figure 3b-3e? As above.

---

## Author Comment (AC2)

**GENERAL COMMENTS:**

The manuscript by Martin Austin et al. presents new observations of the hydrodynamic conditions inside a tidal wake behind an offshore wind turbine monopile in comparison to undisturbed conditions. The measurements took place at an offshore wind farm in the Liverpool Bay and provide valuable insights into the turbulence production caused by the underwater drag of wind turbine installations.

The manuscript is well written and follows a clear storyline. The data of the study are novel and present new results that are important for the understanding of the hydrodynamic footprint of offshore wind farms in shelf seas. However, I recommend minor revisions before acceptance of this manuscript. Please find my comments below.

We thank the reviewer for their support of our manuscript and their suggested minor revisions. Below we outline how we have incorporated revisions into the manuscript and provide some further commentary on their specific questions/suggestions.

In addition to the specific comments, I have a general recommendation for the overall storyline of the manuscript (not mandatory):

- This study is of much importance for the scientific community dealing with offshore wind farm effects. However, to me it reads like a "fieldwork-bubble" paper. I think it could be of great value, if you think outside of your personal research bubble and try to present your research in a less "bubble-specific" manner to attract more readers to this obviously intriguing topic. Explain technical aspects a bit more detailed for non-fieldworkers and elaborate more on the implications of your findings for hydrodynamics, nutrient transport, shelf-sea modeling, etc... Try to think outside of your research area to help solving the big puzzle of offshore wind farm effects on shelf seas and how to study them properly. This is a useful comment, but one that we don't feel able to fully address within the present manuscript. We believe that our present work based on detailed field observations provides the necessary fundamental first steps of quantifying the magnitude and scale of the wake effects prior to up-scaling with numerical models to explore their impacts. Sections 4.2 and 4.3 discuss our findings with respect to the broader scale, for example describing how enhanced seabed stress may transition a live-bed regime towards one which supports bedform development driving a change in seabed habitat, and how the changes to the vertical mixing of the water column in shelf-scale models will need to reflect wakes.

- You present a comprehensive description of your measurements and the impact of monopile mixing, but lack a bit of interpretations of your findings, which can become very important for the process understanding of wake turbulence and the transfer/applications in other areas or studies. Why are these changes in the wake emerging as they do? Why do you see less impact near the bottom in a well-mixed water column? What do you expect for stratified deeper water columns with regard to vertical velocity profiles, turbulence and viscosity? Do you still expect no effect in the lower part of the water column? Which role do the tides and associated bottom mixing

play? What will be the difference between floating and fixed turbines? The reviewer raises some important points here, but we feel that to fully address them is beyond the scope of this manuscript. We feel that the *"...changes in the wake emerging as they do? [...] Why do you see less impact near the bottom in a well-mixed water column?"* are already addressed in sections 3.3, 4.1.

The transition to stratified deeper water columns presents a much greater challenge, since at present there is a deficit of studies addressing wakes in stratified fluids. We refer to Dorrell et al. (2022), where we have some over-lapping co-authorship, who explore in detail the difficulties of expanding into stratified waters to explore *"What do you expect for stratified deeper water columns with regard to vertical velocity profiles, turbulence and viscosity? Do you still expect no effect in the lower part of the water column?"* To achieve those advances, field observations must be linked with high resolution LES modelling to describe the turbulent kinetic energy budget that includes the advected turbulence combining barotropic and baroclinic processes – we include comment to this effect in lines 30 – 34, 242 – 244, and 254 – 256.

**SPECIFIC COMMENTS:**

- Just a suggestion: how about using "tidal wake" instead of "near wake" in the title? Good suggestion, thank you – title updated.

- L1: The first sentence in the abstract is quite lengthy. Consider to split it into two sentences. Done.

- L7: Can you specify "above mid-depth"? Is it around the pycnocline? In water depths less affected by tidal-induced bottom mixing? There's no pycnocline due to the well-mixed conditions and the large tidal range means the water depth is varying between 12.5 and 21 m during spring tides. We have adjusted this to state: "...in the upper half of the water column, ...".

- L10: Your research and title specifically address the impact on bed shear stress, however here you mention floating wind turbines and their potential impact on stratified waters. This sounds like contradictory messages to me. Do floating wind turbines reach down to the bottom and affect the seabed? Don't we expect primarily impact on surface waters for future floating wind turbines? Maybe remove "floating" here and focus on the impact of monopile mixing in stratified waters, as you can find them for example in some areas of the North Sea. We're reporting impacts on both bed sheer stress and water column mixing, which will both occur, but to different degrees, with bottom-fixed and floating structures. Floating turbines will primarily impact the water column, but the current Hywind Scotland floating turbine spar buoys are 78m deep extending to within 17m of the seabed so may impact the benthic boundary layer. Their mooring structures also generate wakes at the seabed. We have adjusted the text

to add: *"…expand into deeper seasonally stratified waters using bottom-fixed and floating structures, …".*

- L17: "45 GW". Are you sure about this number? This seems very high to me for only the northwest European waters. According to my sources entire Europe had about 272 GW of installed wind power capacity in 2023, with 238 GW onshore and 34 GW offshore. Please check this again. We originally included the capacity from the new Seagreen and Dogger Bank A, B and C windfarms for the 45 GW figure, but as Dogger Bank is still not fully operational, we have reverted to 36 GW.

- L26/27: "wakes being efficiently eroded in highly turbulent environments". Do you have evidence/sources for this? The temporal and spatial persistence of wakes is (partially) a function of the degree of background turbulence, with the wake decaying faster in regions of higher turbulence, i.e. close to the seabed. See for example: Eames, I., Jonsson, C., & Johnson, P. B. (2011). The growth of a cylinder wake in turbulent flow. Journal of Turbulence, 12(39), 1–16. https://doi.org/10.1080/14685248.2011.619985

- L30: "adding turbulent energy to the pycnocline". Do you know how deep floating structures reach? If so, add values and sources. We have added depths and sources for the Hywind Scotland floating turbines.

- L32: Please check this sentence again. Is there a word missing after (Schultze et al., 2020)? Thank you, an "and" was missing.

- L36/37: "We use field observations in a tidally energetic well-mixed environment". Didn't you say in Line 26 that wakes are eroded in turbulent environments?! Why do you and how can you observe/study wakes in tidally-energetic and well-mixed environments then?! Can we consider your measurements appropriate/meaningful? This work was originally motivated as part of a much larger project that quantifies the role of fluid-monopile interactions in driving seabed change and impacting ecosystem dynamics. The impact of monopile wakes is readily observable on the seabed where we see changes to bedform dimensions (e.g., Couldrey et al., 2020) reflecting changes to the sediment transport regime. In tidally energetic environments, high flow velocities result in well-mixed water columns and generate strong tidal wakes past structures, but the high levels of turbulence generated (for example) by flow over the seabed is one factor responsible for the spatial and temporal decay of that wake (e.g., Eames et al., 2011). We are therefore observing a strong, well-developed tidal wake from a monopile in shallow waters, where there are resultant impacts on the seabed and within the water column, and we are investigating several factors that are responsible for how this wake evolves in space and time.

- L40: Please add a short introduction of your study area for readers who are not familiar with the UK waters and particularly the Liverpool Bay (where are we, what are the typical/hydrographic conditions, which season are we looking at in this study, when did the measurements take place?). Also add this to Figure 1! (see my comments on

Figure 1 below) Done. We have reorganised section 2, with the new section 2.1 introducing Liverpool Bay and 2.2 the specifics of the monopile wake measurements.

- L42: "454337.8 mE, 5915789.1 mN, UTM30U". Please use latitude/longitude coordinates, which makes it much easier to read and to locate your research site. Adjust in Figure 1 accordingly. Done.

- L47: Is velocity during ebb tides weaker due to drag of turbine? Velocity has been measured inside wake, hasn't it? Yes, the drag of the monopile does weaken the ebb flow (cf. Fig.5a), but we compare this to our other nearby observations in Fig 2a and show that the ebb tides are weaker away from the influence of the monopile. Also refer to the cited publication Unsworth et al. (2023).

- L50: What value did you use for the kinematic viscosity in the calculations of the Reynolds number? Added to the text at line 56 (1.36E-06 $m^2s^{-1}$).

- L51: Provide a source for "The transition to the turbulent critical flow regime begins at Re…". We've added a citation to Williamson (1966).

- L52: Explain in more detail when and why we observe Karman vortex streets behind a cyclinder. This comes a bit out of the blue here. We have added a little more detail at line 57 – 61 plus Fig.2 caption and two citations to the literature, to highlight the form that we expect the wake to take during the measurement period.

- L55: Use lat/lon coordinates. Done

- L58/59: "timeseries of surface and bottom temperatures recorded nearby during 2018 indicate that thermal stratification only occurs over neap tides during Jun – Jul". What about interannual variability? Do you have any long-term evidence for this assumption? Are you sure stratification is only occurring during this period? What are the conditions during your measurements? We have added citations to Rippeth et al. (2001) and Simpson et al. (2002) which provide background on our assertion that the water column is well-mixed and not stratified, and link to the substantial body of long-term evidence that describe the dynamics of the eastern Irish Sea and Liverpool Bay.

- L59: "Jun-Jul". You're saving just on letter each... please write June-July. Done

- L61: "Nortek Signature 1000 5-beam ADCP". As I am not familiar with measurement devices, could you please explain here shortly what is special about this device and what/how it measures? As mentioned before, make this paper accessible to readers outside of "fieldwork bubble". Same accounts for "pitch and roll" in L63. This is a modern acoustic doppler current profiler that is well-used for oceanography. We have added to the definition of ADCP, but feel any further explanation is unnecessary as the details and merits are easily accessible in both general form via an internet search and in numerous peer-reviewed publications.

- L110-118: Can you explain why this happens? What is the role of the wake here? We have modified the text to include a little more context: *"Following similarity theory, a local balance should exist between \varepsilon and P based on a constant stress relationship, with the local production of turbulence by tidal shear at the seabed [...] During downstream conditions, \varepsilon_5 is one and 1.5 orders of magnitude greater than predicted at the seabed and near-surface, respectively, and peaks prior to the occurrence of the strongest flows (mid-ebb); this likely reflects the addition of wake turbulence to the water column."*

- L129-132: In addition to your explanations, could it be that the generated turbulence inside the wake causes the water column to become more uniform (erode vertical density gradients), which results in a more uniform velocity profile as you observe it here? We consider this highly unlikely in the present setting – the water column is well-mixed and there are no vertical density gradients, plus the downstream (wake-effected) velocity profile is not uniform (cf. Fig. 4f and Fig. 5a). It is probable that in the lower region of the water column the turbulence generated by shear at the seabed is eroding the wake and forcing a recovery of the flow back towards a classical benthic boundary layer structure as we discuss at line 151 – 152.

- L135-150: Please add (4a), (4b), (4c) more often to make it easier to follow your analysis. Further reference to the figure panels has been added.

- L135-151: Can you relate the surface-to-bottom patterns you observe in (4a-c) to the wake processes? At this location in the manuscript we are describing the contrasting pattern of the observations between the wake and background flows, rather than interpreting their surface-to-bottom distribution, but we have made a revision to the text (line 150 – 156) to highlight to the reader that we are focusing on the premise that the background flows are dominated by tidal shear at the seabed, whereas the wake flow displays significant variations at all depths: *"During the upstream phase (flood), the mean velocity profile displays a logarithmic form and extends the full thickness of the water column as expected in a system dominated by tidal shear at the seabed. The maximum values of the stress occur at or just above the seabed and tend towards zero at the surface \citep{Rippeth2002}; an increase in stress is observed for z/h > 0.65, which may be due to surface waves. Following \citet{Rippeth2003}, for our observed wave amplitude and period of ~0.5 m and ~4 s, respectively, we may expect a bias in the stress term close to the surface of O(0.5) N m^{-2} decreasing with depth to O(0.05) N m^{-2} at the bed, which is in close agreement with our observations."*

- L153: "... for sediment studies". Specify sources here. We have added a citation to support this statement.

- L160-162, Figure 4: For the steady flow model, which value are you using for u_star? Shouldn't you have two reference lines, one for flood (u=0.8m/s) and one for ebb (u=0.6m/s)? I might be wrong here. We have added an additional equation (Eq. 2) to the manuscript to clarify this point. The theoretical form of $N_z$ using a steady flow model takes a parabolic form where the stress decreases linearly from its maximum value at the seabed to zero at the sea surface and the shear is controlled by the logarithmic lawof-the-wall relationship. When $N_z$ is normalised with ($\kappa u_* h$), to give $Nz'$ as plotted in Figure 5, $u_*$ cancels, so Fig. 5e is correct with only one reference line.

- L165/166: "rapidly increases in the upper water column where we observe the strong deficit in the mean velocity.". Don't we observe the same increase at 0.6 for the flood tide? Both the background (flood) and wake-effected (ebb) periods display elevated Nz above z/h=0.6, but within the wake this increase is observed to begin from much lower in the water column (z/h ~0.35). This corresponds to the elevations in Fig.5b and c, where we also observe increases in stress and dissipation rate. We have updated the text for clarity (line 178 – 184): *"We compare N_z computed for the ensemble-averaged peak upstream and downstream tidal phases in (Fig. 5(e)). During the upstream phase, although there is some significant deviation from the steady-flow model approaching the sea surface for z/h > 0.6, the observations are consistent with the model in both trend and magnitude. However, during the wake-effected ebb tide, N_z is almost an order of magnitude larger than the steady-flow model and although approximating a parabolic form at z/h < 0.35, it rapidly increases in the upper water column between z/h = 0.35 and z/h = 0.6; this corresponds to the region where shear, stress and \varepsilon_5 are observed to increase for the wake-affected flow in Fig. 5(a--c) and suggests that the monopile drives enhanced vertical mixing through the water column.".*

- L168 4.1 Wake Enhanced Turbulence: Coming back to "think outside the box", can you somehow quantify the increase in turbulence/TKE that you observe inside the wake? This would be of great value for, e.g. shelf sea models, which need to parameterize the additional turbulence increase (as you mention in L200 as well). This is an insightful comment that we would ultimately agree with, but in the present field-based study we are reporting on measurements made at one distance from a monopile approximately in the centreline of the wake. To quantify the increase in turbulence we need to know both the length and width of the wake as it propagates and evolves downstream from the monopile. We are addressing this question in other areas of our work by linking 3D numerical modelling to our field observations.

- L184-187: Is this also expected close to the pile or only in distances ~40 m, where the wake is already weakened? This is an important question that is the focus of our ongoing work where we address the length-scale of the wake by combining field observations with 3D numerical modelling. The wake will only be fully established a couple of monopile diameters downstream of the monopile, where the flow converges after separating around the structure, and then we expect the wake to weaken and expand horizontally moving downstream as the mean flow is restored and the wake becomes indiscernible from the background flow. We therefore expect the processes responsible for eroding the wake to be consistent along its entire length, but to change in magnitude.

- L197: Rephrase. Don't start sentence with "C_d...". Changed

- L199: "but this more than doubled". Is there a word missing? Thank you – corrected.

- L201: Add that such shelf-scale models must account for the additional turbulence, e.g. via a drag parameterization. Examples of such parameterization approaches can be found in Rennau et al. 2012 and Christiansen et al., 2023. We have added citations to these additional references.

- L202: I'm a bit confused, why is larger bottom drag linked to enhanced sediment transport? Don't we expect slower velocities for larger drag and thus less sediment transport in the lee of monopiles? Or is this related to turbulence and resuspension. The drag from the monopile reduces the mean flow and transfers momentum to the turbulent regime. This increases the stress acting on the seabed and may enhance sediment transport. We are specifically referring to the seabed drag coefficient that parameterises this transfer of momentum from the water column to the seabed (Fig. 5d) and thus forces sediment transport using concepts such as the quadratic stress law. We have modified the text (line 219 – 221) to clarify this point for the reader. *"Unless this wake-enhanced transfer of momentum from the mean flow to turbulence is accounted for, shelf-scale models (e.g., Telemac, FVCOM) that generally use the quadratic stress law as the default method of estimating bed shear stress will under predict seabed stress".*

- L224 4.4 Broader Consequences of Enhanced Mixing: How do you expect your presented impacts on velocity, TKE and viscosity in an stratified water column compared to a well-mixed water column? Same trends in the vertical distributions? These are important impacts that we have debated including here in greater detail, but we have deliberately chosen to limit our discussion of them due to a lack of understanding of wake dynamics in stratified fluid flows. The fundamental change is the addition of baroclinic to the barotropic drag due to the vertical density interface and the advection of wake turbulence. This is an area that requires closely linked physical measurements, large eddy simulation modelling and parameterisation for larger-scale models. We have modified our text to state this and feel that section 4.4 provides a broad summary of the transition towards stratified waters based on synthesising current knowledge, whilst suggesting a path forwards.

- Figure 1: I think there is an overload of information on just two panels and still geographical information is missing in my opinion. Please revise this figure to highlight all these important information more clearly. (1.) Add another panel/inset to show the geographical area we are looking at. Zoom out to show the Irish Sea and eastern UK coastlines for readers who are not familiar with Liverpool bay. (2.) Indicate the location of the offshore wind farm on your map, e.g. by drawing a polygon. This does not become clear by just looking at the bathymetry. (3.) Plot both measurement stations on the large panel and use the same color (red). (4.) Use lat/lon coordinates. (5.) My suggestion: Instead of using one large panel for the map with an inset for the zoom, split this into three sub-panels showing (a) the geographical region, (b) the zoom on Liverpool bay with the polygons and markers for the wind farm location and measurement stations (bathymetry in the back), and (c) the zoom onto the single turbine plus the measurement station. You could arrange this by plotting the "geo region" panel on the left and the two "zoom" panels on the right (top and bottom), which gives you a nice overview about all three panels and all important geo

information. You could add your panel (b) below as the new panel (d), or to take some load of this figure, plot it in an additional figure, where you plot (a) the velocities and (b) the histogram. This would also enlarge the histogram, which makes it easier to read and easier for you to indicate the critical Reynolds regimes. Also see my drawings in the attached PDF. We thank the reviewer for their suggestion for a revised version of Figure 1. We have modified the figure, broadly as suggested, and feel that it now provides greater clarity for the reader.

- Figure 1 Caption: remove "..., and indicating the flood tide direction.". This does not really become clear to me by just looking at the bathymetry and is better shown in b). Changed.

- Figure 2 and 3: Why are you using contour lines and not continuous colors? I would recommend to use continuous colorscales. Especially in Figure 2, contour lines overshadow the actual colors. If you are using contours regardless, please indicate them in the colorbar. We use contours to represent the discrete nature of the variables that are plotted in log-space, and this is reflected in the discrete colorbar scaling.

- Figure 2 Caption: Write names for dissipation and production in (b) and (c) not only their special characters. We have added the full variable names.

- Figure 4: Describe the small inset plotted in (a). Explain "df" used in (d). Description added.

[Figure]

Figure 2:

[Figure]

a)

N

E

b)

n

Re_critical

Red